# Low-Protein Diet Supplemented with Medium-Chain Fatty Acid Glycerides Improves the Growth Performance and Intestinal Function in Post-Weaning Piglets

**DOI:** 10.3390/ani10101852

**Published:** 2020-10-12

**Authors:** Zhijuan Cui, Xianze Wang, Zhenping Hou, Simeng Liao, Ming Qi, Andong Zha, Zhe Yang, Gang Zuo, Peng Liao, Yuguang Chen, Bie Tan

**Affiliations:** 1College of Animal Science and Technology, Hunan Agricultural University, Changsha 410128, China; czj123@stu.hunau.edu.cn (Z.C.); wxz13187058389@163.com (X.W.); dysbio@163.com (G.Z.); liaopeng@isa.ac.cn (P.L.); 2Laboratory of Animal Nutritional Physiology and Metabolic Process, Key Laboratory of Agro-ecological Processes in Subtropical Region, National Engineering Laboratory for Pollution Control and Waste Utilization in Livestock and Poultry Production, Institute of Subtropical Agriculture, Chinese Academy of Sciences, Changsha 410125, China; lsm19931@163.com (S.L.); qmcharisma@sina.com (M.Q.); zhaandong18@mails.ucas.ac.cn (A.Z.); promise.yangzhe@foxmail.com (Z.Y.); 3Institute of Bast Fiber Crops, Chinese Academy of Agricultural Sciences, Changsha 410205, China; houzhenping@caas.cn

**Keywords:** medium-chain fatty acid glycerides, intestinal permeability, intestinal morphological structure, tight junction protein, cytokines

## Abstract

**Simple Summary:**

After weaning, piglets cannot absorb protein well and cannot get enough energy from the diet due to intestinal dysplasia. Medium-chain fatty acids are very effective in providing energy for piglets and may protect the integrity of the intestinal barrier to improve the healthy development of piglets. Therefore, we speculate that medium chain fatty acid glycerides can promote the growth of weaned piglets in a low protein diet. The present study examined the effects of medium-chain fatty acid glycerides on the growth performance, intestinal barrier function and inflammatory response of weaned piglets. These findings provide a new prospect for the application of medium-chain fatty acid triglycerides in piglets.

**Abstract:**

Medium-chain fatty acid glycerides have been shown to provide energy for rapid oxidation in the body. The study was conducted to investigate the effects of dietary supplementation with medium-chain fatty acid glyceride on the growth performance and intestinal health of weaned piglets fed with a low-protein diet. Nighty healthy weaned piglets were randomly divided into five treatments: NP (Normal protein treatment, normal-protein diet no antibiotics included); NC (Negative control, low-protein diet no antibiotics included); PC (Positive control, low-protein diet +75 mg/kg quinocetone, 20 mg/kg virginiamycin and 50 mg/kg aureomycin); MCT (tricaprylin + tricaprin treatment, low-protein diet + tricaprylin + tricaprin); GML (glycerol monolaurate treatment, low-protein diet + glycerol monolaurate). The results showed that the average daily feed intake (ADFI) of the MCT treatment was significantly higher than that of the NP, NC treatments (*p* < 0.05). In the jejunum, the villus height of the GML treatment was significantly lower than that of the PC treatment (*p* < 0.05), and the number of goblet cells in the GML treatment was higher than that in the NC treatment (*p* < 0.05). Compared with the NC treatment, the MCT treatment significantly increased the level of claudin-1, Zonula occludens-1(ZO-1), while the GML treatment significantly increased the level of claudin-1, occludin, ZO-1 (*p* < 0.05). In the ileum, the level of ZO-1 in the GML treatment was significantly higher than that in the NP, NC, PC treatments (*p* < 0.05). Compared with the NC treatment, the GML treatment significantly increased the level of Secretory immunoglobulin A (SIgA) in the ileum and serum, while the MCT treatment significantly increased the level of SIgA and decreased the level of interleukin-6 (IL-6) in the ileum (*p* < 0.05). These results showed that the addition of medium-chain fatty acid glycerides to a low-protein diet could improve the growth performance and intestinal functional barrier of weaned piglets and also improve the immune function of weaned piglets.

## 1. Introduction

The intestinal mucosa of piglets has notably high energy demands because of their physiological functions, including the absorption and transport of nutrients, signal transduction, as well as the renewal of epithelial cells and maintenance of the structure [1]. Furthermore, piglets can make good use of breast milk to provide energy when they are not weaned, because they have a high lipase activity from the moment of birth [2]. However, its nutritional supply changes from high digestibility breast milk rich in protein, fat and lactose to a starch-based diet with lower digestibility after weaning [3,4]. Meanwhile the inadequate feed intake immediately after weaning results in an energy intake that is insufficient for maintaining the structure and function of the intestinal epithelium [5]. A previous study showed that early weaning downregulated intestinal epithelial energy production in piglets, including the tricarboxylic acid cycle pathway, fatty acid oxidation and glycolysis [6]. Thus, a normal mucosal energy status is an important guarantee for relieving weaning stress and improving the growth performance of piglets [7].

Medium-chain fatty acid glyceride is an ester formed from glycerol and fatty acid. Because of its small molecular weight, small volume and strong water solubility, it is easier to absorb. Medium-chain fatty acid glycerides are decomposed into medium-chain fatty acids (MCFAs) in the small intestine, and can then quickly enter the liver through the portal vein for oxidation to provide energy.

Furthermore, compared with long-chain fatty acids (LCFAs), the oxidation of medium-chain fatty acids is independent of the carnitine system and can be activated directly in the mitochondrial matrix [8]. The unoxidized MCFAs could synthesize triglyceride (TG) by synthesizing LCFAs, after which TG was transported to adipose tissue for storage by forming chylomicron [9]. According to Dierick et al., medium-chain triacylglycerols have a good antibacterial effect [10,11]. Due to the change of feed during weaning, many undigested nutrients in piglets provide a suitable hotbed for the reproduction and development of bacteria [12]. Caprylic acid and lauric acid have ruined the cytoplasmic structure of bacteria, and caprylic have significantly reduced the number of *Escherichia coli* and *salmonella* [13].

Because of the incomplete intestinal development of piglets, piglets are sensitive to the protein level in diets. The level of protein is an important factor affecting the growth performance and intestinal health of piglets [14,15]. The undigested protein of piglets will enter the large intestine for fermentation in order to produce toxic metabolites, which will destroy the integrity of the intestinal epithelium and damage the health of the intestinal tract [16]. Many studies have shown that reducing protein levels can reduce the diarrhea rate and N excretion of piglets, and can reduce protein fermentation in the intestinal tract [17,18].

Therefore, the purpose of this study was to investigate the effects of dietary supplementation with medium-chain fatty acid glyceride on the growth performance and intestinal health of weaned piglets, based on a low-protein diet that was beneficial for relieving the nutritional burden [19,20]. In this experiment, 21-day-old weaned piglets were fed with two kinds of medium-chain fatty acid triglycerides at a low protein diet level, and the antibiotic group and normal protein group were compared at the same time. The growth performance, intestinal permeability, intestinal morphology, intestinal tight junction proteins and intestinal cytokines of piglets were detected to provide a scientific basis for the application of medium-chain fatty acid triglycerides in low-protein diets.

## 2. Materials and Methods 

This study was conducted in accordance with the guidelines of the Institute of Subtropical Agriculture, Chinese Academy of Sciences. All the experimental schemes were approved by the Animal Ethics Committee of the Institute of Subtropical Agriculture, Chinese Academy of Sciences. The animal experiments project identification code is IACUC#20190615, and the approval date is 15 May 2019.

### 2.1. Experimental Animal and Sample Collection

A total of 90 healthy Duroc × Landrace × Large Yorkshire piglets weaned at 21 days of age (body weight 6 ± 0.15 kg) were randomly assigned to five treatments, with six pens per treatment and three piglets per pen. The study design is shown in Table 1: NP (Normal protein treatment, normal-protein diet no antibiotics included); NC (Negative control, low-protein diet no antibiotics included); PC (Positive control, low-protein diet +75 mg/kg quinocetone, 20 mg/kg virginiamycin and 50 mg/kg aureomycin); MCT (low-protein diet + tricaprylin/tricaprin); GML (low-protein diet +glycerol monolaurate). The normal protein basal diet and low-protein basal diet were formulated according to the nutrient requirements for weanling piglets (NRC, 2012) and previous studies [21] (Table 2). The medium-chain fatty acid glycerides were obtained from Deyuanshun Biological Technology Co., Ltd. (Beijing, China). The experiment lasted for 14 days, and all piglets were freely fed.

On the second morning after the 14th day, we collected the blood and tissue. After overnight fasting, we collected blood from the jugular vein of piglets in the morning. Approximately 10 mL of blood from the jugular vein was collected in aseptic capped tubes containing 150 U of sodium heparin and an ordinary centrifuge tube. Serum and plasma samples were obtained by centrifugation at 3000× *g* for 10 min at 4 ℃ and were stored at −80 ℃ for biochemical detection. Six piglets from each treatment were anesthetized with sodium pentobarbital (20 mg/kg BW) and killed by jugular puncture. After slaughtering, the samples of jejunum and ileum were immediately snap-frozen in liquid nitrogen and then transferred to −80 ℃ for further analysis. The jejunum and ileum (around 2 cm) was fixed in 4% formalin to detect the morphology of the intestine. The chyme of the colon was collected in a 50 mL centrifuge tube and then transferred to −80 ℃ for a fatty acid determination.

### 2.2. Serum Biochemical Indexes Assays

Biochemical indicators (D-LACT: mmol/L, DAO: mmol/L) were measured using an instrument (Biochemical Analytical Instrument, Beckman CX4, Beckman Coulter Inc., Brea, CA, USA) and commercial kits (Sino-German Beijing Leadman Biotech Ltd., Beijing, China).

### 2.3. Detection of Intestinal Morphology and Structure

HE staining was used to detect the morphology of the jejunum and ileum. The jejunum and ileum samples were cut open and dehydrated in a tissue-processing machine for 16 h before embedding paraffin. Each section was cut into a thickness of about 4 um, and was then fixed in a glass slide and heated at 57 ℃ until the sample was dry. The sample was stained with hematoxylin and eosin and loaded with cover slides. The villus height and crypt depth of the jejunum and ileum were measured by microscope.

### 2.4. Determination of Cytokines in the Serum and Intestine

The serum, jejunum and ileum concentrations of SIgA, interleukin-1 beta (IL-1β), interleukin-6 (IL-6), tumor necrosis factor-alpha (TNF-α) and interferon-gamma (IFN-γ) were measured using commercially available swine enzyme-linked immunosorbent assay (ELISA) kits, according to the manufacturer’s instructions (Meimian Industrial Co., Ltd., Nanjing, Jiangsu, China).

### 2.5. Immunohistochemical Analysis

The protein expressions of claudin-1, occluding and ZO-1 in the jejunum and ileum of piglets were determined using an immunohistochemical analysis. The tissue blocks were fixed with 14% paraformaldehyde, and then continuous paraffin sections were made. The paraffin sections were baked in an oven at 65 ℃ for 2 h, washed with PBS three times for 5 min each time, before being repaired by microwave in ethylenediamine tetraacetic acid (EDTA) buffer. And after the medium heat boils, the power is cut off, and then after 10 min, it boils on a low heat. After this, they were washed three times with PBS, as before, before being put in a 3% hydrogen peroxide solution, incubated 10 min away from light at room temperature, washed with PBS three times for 5 min each time, and sealed with 5% Bovine Serum Albumin (BSA) for 20 min after drying. Then, we removed the BSA solution, added about 50 μL diluted LRP6 covering tissue to each slice, which were then left overnight at 4 ℃. After this, they were washed with Phosphate Buffer Saline (PBS) three times, and 50–100 μL of the corresponding species of HRP labeled goat anti-rabbit (purchased from BOSTER company) was added to the slices, which were incubated at 37 ℃ for 50 min. They were PBS-washed three times again, and, except for the PBS solution, 50–100 μL freshly prepared DAB solution was added to each slice, followed by a microscope-controlled color development. After the complete color development, they were rinsed with distilled water or tap water, followed by hematoxylin redyeing, 1% hydrochloric acid alcohol differentiation (about 1 s), a tap water rinse, ammonia return to blue, and a running water rinse. The slices were treated with gradient alcohol for 10 min, dehydrated and dried, transparent xylene was used, and the slices were sealed with neutral gum. Staining sections were independently reviewed and scored by two researchers using a 400-fold magnification microscope (Olympus, Tokyo, Japan).

### 2.6. Statistical Analyses

Data were analyzed by an analysis of variance, using the General Linear Models procedure of the SPSS 20.0 (SPSS Inc., Chicago, IL, USA). Significant differences between means were determined using the Tukey’s multiple comparison tests. Results were expressed as the mean ± standard error of the mean (SEM). A value of *p* < 0.05 was considered statistically significant. Mean values and a statistical elaboration were performed by using each pen as the experimental unit (*n* = 6 per treatment).

## 3. Results

### 3.1. Growth Performance

The average daily gain (ADG), average daily food intake (ADFI) and feed/gain ratio of piglets (F/G) are shown in Table 3. There were no differences in ADG and F/G among all treatments (*p* > 0.05). The MCT treatment significantly increased ADFI when compared with the piglets of the NC and NP treatments (*p* < 0.05), but there were no differences in ADFI among the PC, MCT or GML treatments (*p* > 0.05).

### 3.2. Intestinal Permeability

Table 4 shows the indications of the intestinal permeability, serum concentrations of D-LACT and DAO, of piglets. The concentration of D-LACT in the serum of piglets in the MCT treatment was lower than for the NP treatment (*p* < 0.05), but the MCT and GML treatments did not affect the D-LACT concentration when compared with the NC and PC treatments (*p* > 0.05). 

### 3.3. Intestinal Morphology

The results of the lymphocyte and goblet cell numbers, as well as the villus height and crypt depth of the jejunum and ileum, are shown in Figure 1 and Figure 2. In the jejunum, the goblet cell numbers in piglets of the GML treatment were higher than those of the NC-treated piglets (*p* < 0.05), and there were no differences among piglets of the NP, PC, MCT and GML treatments (*p* > 0.05). The villus height was significantly decreased in the piglets of the GML treatment when compared with the PC treatment (*p* < 0.05), but there were no differences among piglets of the NP, NC, MCT and GML treatments (*p* > 0.05) (Figure 1). In the ileum, there were no difference in the numbers of lymphocytes and goblet cells, as well as in the villus height and crypt depth, among all the piglets from all the treatments (*p* > 0.05) (Figure 2).

### 3.4. The Expressions of Tight Junction Proteins in the Small Intestine of Piglets

Figure 3 and Figure 4 show the protein expressions of claudin-1, occludin and ZO-1 in the jejunum and ileum of piglets. In the jejunum, the claudin-1 protein expression of the MCT treatment and GML treatment was significantly lower than that of the NP treatment, but significantly higher than that of the NC treatment (*p* < 0.05). Furthermore, the GML treatment significantly increased the claudin-1 protein expression when compared with the PC treatment (*p* < 0.05) (Figure 3A). The protein expression of occludin in piglets of the GML treatment was significantly higher than that of the NP, NC, PC treatments (*p* < 0.05) (Figure 3B). The MCT and GML addition significantly increased the ZO-1 protein expression when compared with that of the NP and NC treatments (*p* < 0.05) (Figure 3C).

In the ileum, there were no differences in the claudin-1 protein expression among the piglets of the NP, NC, MCT and GML treatments (*p* > 0.05), but its expression in piglets of the PC treatment was higher than for the other four treatments (*p* < 0.05) (Figure 4A). The protein expressions of occludin were significantly decreased in piglets of the MCT and GML treatments when compared with the NP treatment (*p* < 0.05) but were not affected when compared with the NC and PC treatments (*p* > 0.05) (Figure 4B). The expression of ZO-1 protein in piglets of the GML treatment was significantly higher than that in the NP, NC treatments, and MCT addition also significantly increased the ZO-1 protein expression when compared with the NC treatment (*p* < 0.05) (Figure 4C).

### 3.5. The Concentrations of Cytokines in the Small Intestinal Mucosa and Serum 

In the jejunal mucosa, there was no significant difference in IL-1β and IL-6, IFN-γ among the treatments. The MCT and GML treatments increased the concentrations of SIgA in piglets when compared with the PC treatment and decreased the concentrations of TNF-α in piglets when compared with the NP treatment (*p* < 0.05) (Table 5). 

In the ileal mucosa, when compared with the NC treatment, the piglets in the MCT and GML treatments showed an increase in the SIgA concentration, but a decrease in the concentrations of IL-6 and TNF-α (*p* < 0.05). Additionally, the dietary supplementation with MCT and GML increased (*p* < 0.05) the SIgA concentration but did not affect (*p* > 0.05) the concentrations of inflammatory cytokines (IL-1 β, IL-6, TNF-α, IFN-γ) when compared with the PC treatment (Table 6). 

In the serum, the SIgA concentration in piglets of the GML treatment was significantly higher than that of the NC and PC treatments (*p* < 0.05). MCT addition decreased the IL-6 concentration when compared with the PC treatment, but increased its concentration when compared with the NP treatment (*p* < 0.05) (Table 7).

## 4. Discussion

The present study demonstrates the promotion of the intestinal health of medium-chain fatty acids, which is key to maximizing growth performance in livestock. In our study, dietary supplementation with medium-chain fatty acid glycerides in a low-protein and antibiotic-free diet improved the growth performance, intestinal barrier and inflammatory reaction of weaning piglets.

First, the addition of medium-chain fatty acid glycerides in diets might inhibit the increase of the intestinal mucosal permeability of piglets caused by weaning stress. DAO is closely related to the maturity and integrity of intestinal mucosal cells [22,23], and D-LACT can enter the bloodstream through the damaged mucosa [24]. Both of them can be used as an index to detect intestinal permeability. In our study, the D-LACT of serum decreased in the MCT treatment, which may indicate that the intestinal permeability decreased in the piglets. This is consistent with previous studies that showed that MCT could reduce the permeability of CA in Caco-2 cells [25] and inhibit the increase in intestinal permeability induced by LPS in mice [26].

Second, medium-chain fatty acid glycerides could promote the intestinal maturation of piglets, reduce the injury of weaning stress and increase the absorption of nutrients in piglets. The epithelial cells near the villous tip have the strongest digestion and absorption ability of nutrients [27], and thus the increase of epithelial cells and the increase of the villus height can effectively promote the absorption of nutrients [28]. However, it is easy to cause the villus height of piglets to decrease during weaning [29,30]. In our study, the treatment with medium-chain fatty acid glycerides helped increase the villus height and the number of goblet cells, which was consistent with a previous study [31]. Therefore, it could possibly promote the maturation of intestinal epithelial cells and villi in order to absorb more nutrients by increasing the number of goblet cells and the height of villi.

Third, it was further proven that medium-chain fatty acid glycerides could effectively protect the integrity of the intestinal barrier, given that the protein expressions of claudin-1, occludin and ZO-1 were significantly increased. Claudin-1, occludin and ZO-1 are important components of the intestinal barrier [32]. The bacteria and toxins can regulate and control the expression of intestinal occludin and ZO-1 by regulating or influencing cytokines and protein kinase C [33]. By adding medium-chain fatty acid glycerides in our study, the claudin-1, occludin and ZO-1 increased significantly, consistent with a previous study that showed that medium-chain fatty acid glycerides could readjust the distribution of occludin and ZO-1 [34]. The reason for this may be that they could effectively prevent bacterial endotoxin and toxic macromolecules from entering the body in order to increase the expression of claudin-1, occludin and ZO-1 proteins, thus protecting the health of the intestinal barrier. 

In addition, medium-chain glycerides can protect intestinal barrier health by reducing the production of proinflammatory factors. Many studies have shown that weaning stress can easily lead to the production of proinflammatory factors in piglets, including IL-1β, IL - 6, TNF-α, IFN-γ [35,36]. SIgA is the first protective barrier of the intestinal epithelium and can protect intestinal epithelial cells from enterotoxins and pathogenic microorganisms [37]. In our study, MCT significantly decreased the level of IL-6 in the ileum when compared with the NC treatment, and both MCT and GML significantly increased the level of SIgA in the ileum when compared with the NC treatment. This agrees with a previous study, according to which medium-chain glycerides could significantly reduce the level of proinflammatory cytokines, such as IL-6, IL-1, and relieve the inflammation of colitis in rats [38,39]. These data show that medium-chain fatty acid glycerides can, to a certain extent, alleviate the intestinal inflammatory response caused by weaning stress. The reduction of inflammation is more conducive to the protection of the intestinal barrier and plays a positive role in the growth of piglets.

## 5. Conclusions

This study shows that the supplementation of medium-chain fatty acid glycerides to a low-protein diet can decrease the intestinal permeability, improve the intestinal morphology and structure, increase some tight junction protein expressions, promote the secretion of IgA, and inhibit the production of inflammatory cytokines in weaning piglets. These findings illustrate that medium-chain glycerides can improve the growth performance of piglets by improving the intestinal barrier and by regulating cytokines.

## Figures and Tables

**Figure 1 animals-10-01852-f001:**
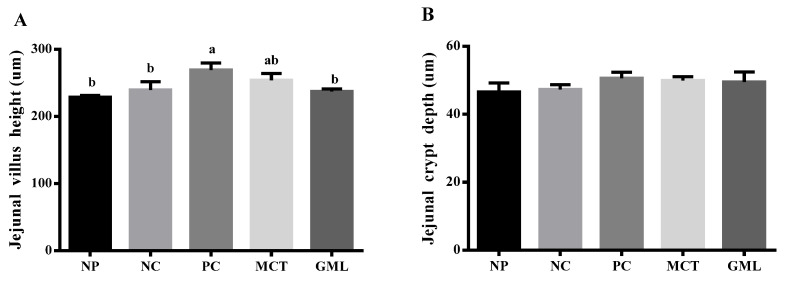
The morphology and structure of the jejunum in the piglets. (**A**): The jejunal villus height. (**B**): The jejunal crypt depth. (**C**): The jejunal lymphocyte numbers. (**D**): The jejunal goblet cell numbers. Values are the mean ± SEM, *n* = 6 per treatment (the *n* = 6 refers to the number of pigs slaughtered and sampled). ^a,b^ Mean values sharing different superscripts within a row differ (*p* < 0.05). NP = normal protein basal diet no antibiotics included; NC = low-protein basal diet no antibiotics included; PC = low-protein basal diet + antibiotics (75 mg/kg quinocetone, 20 mg/kg virginiamycin and 50 mg/kg aureomycin); MCT = low-protein basal diet + 2 kg/T tricaprylin/tricaprin; GML = low-protein basal diet + 2 kg/T glycerol monolaurate.

**Figure 2 animals-10-01852-f002:**
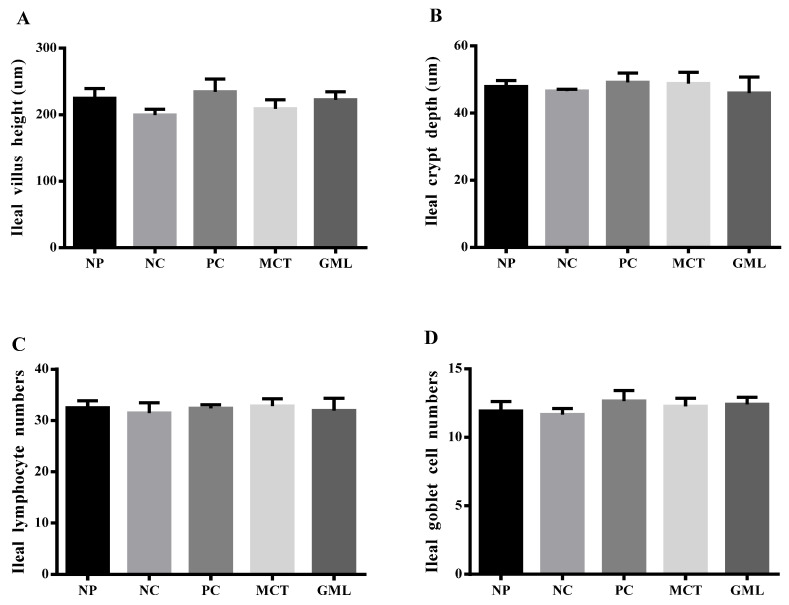
The morphology and structure of the ileum in piglets. (**A**): The jejunal villus height. (**B**): The jejunal crypt depth. (**C**): The jejunal lymphocyte numbers. (**D**): The jejunal goblet cell numbers. Values are the mean ± SEM, *n* = 6 per treatment (the *n* = 6 refers to the number of pigs slaughtered and sampled). NP = normal protein basal diet no antibiotics included; NC = low-protein basal diet no antibiotics included; PC = low-protein basal diet + antibiotics (75 mg/kg quinocetone, 20 mg/kg virginiamycin and 50 mg/kg aureomycin); MCT = low-protein basal diet + 2 kg/T tricaprylin/tricaprin; GML = low-protein basal diet + 2 kg/T glycerol monolaurate.

**Figure 3 animals-10-01852-f003:**
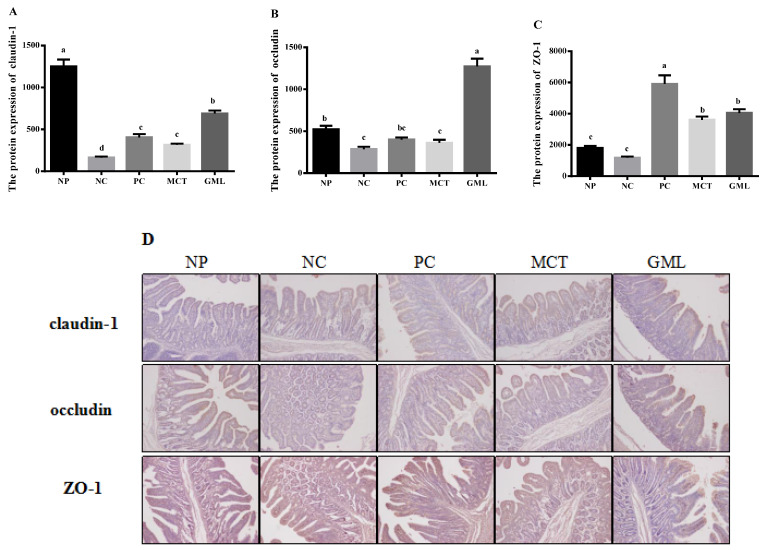
The expressions of tight junction proteins of the jejunum in piglets. (**A**): The protein expression of claudin-1. (**B**): The protein expression of occludin. (**C**): The protein expression of ZO-1. (**D**): The protein expression of claudin-1, occludin, ZO-1. The (**A**–**C**) expressions of tight junction protein and (**D**) representative immunohistochemical staining images in the jejunum of piglets. Values are the mean ± SEM, *n* = 4 per treatment (the *n* = 4 refers to the number of pigs slaughtered and sampled). ^a,b,c,d^ Mean values sharing different superscripts within a row differ (*p* < 0.05). NP = normal protein basal diet no antibiotics included; NC = low-protein basal diet no antibiotics included; PC = low-protein basal diet + antibiotics (75 mg/kg quinocetone, 20 mg/kg virginiamycin and 50 mg/kg aureomycin); MCT = low-protein basal diet + 2 kg/T tricaprylin/tricaprin; GML = low-protein basal diet + 2 kg/T glycerol monolaurate.

**Figure 4 animals-10-01852-f004:**
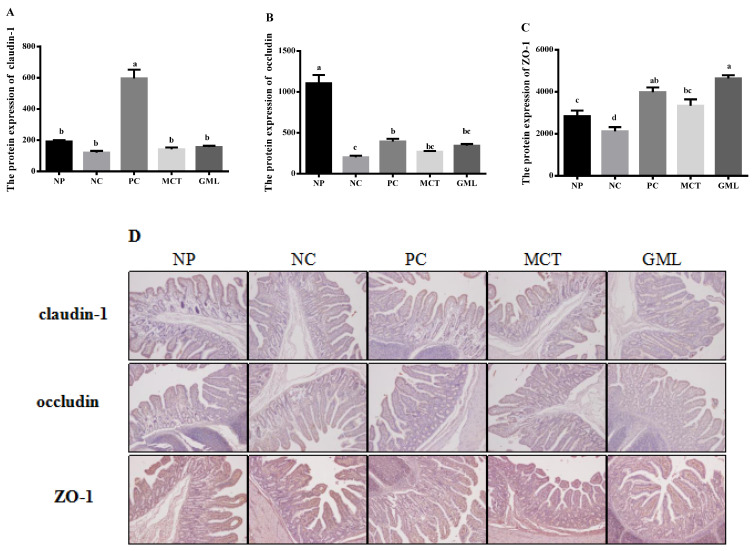
The expressions of tight junction proteins of the ileum in piglets. (**A**): The protein expression of claudin-1. (**B**): The protein expression of occludin. (**C**): The protein expression of ZO-1. (**D**): The protein expression of claudin-1, occludin, ZO-1. The (**A**–**C**) expressions of tight junction protein and (**D**) representative immunohistochemical staining images in the ileum of piglets. Values are the mean ± SEM, *n* = 4 per treatment (the *n* = 4 refers to the number of pigs slaughtered and sampled). ^a,b,c,d^ Mean values sharing different superscripts within a row differ (*p* < 0.05). NP = normal protein basal diet no antibiotics included; NC = low-protein basal diet no antibiotics included; PC = low-protein basal diet + antibiotics (75 mg/kg quinocetone, 20 mg/kg virginiamycin and 50 mg/kg aureomycin); MCT = low-protein basal diet + 2 kg/T tricaprylin/tricaprin; GML = low-protein basal diet + 2 kg/T glycerol monolaurate.

**Table 1 animals-10-01852-t001:** Study design.

Treatments	Description
Normal protein (NP)	normal protein basal diet no antibiotics included
Negative Control (NC)	low-protein basal diet no antibiotics included
Positive control (PC)	low-protein basal diet + antibiotics (quinocetone 75 mg/kg, Virginiamycin 20 mg/kg, Chlortetracycline 50 mg/kg)
Treatment 1 (MCT)	low-protein basal diet + 2 kg/T tricaprylin/tricaprin
Treatment 2 (GML)	low-protein basal diet + 2 kg/T glycerol monolaurate

**Table 2 animals-10-01852-t002:** Composition and nutrient levels of the basal diet (as-fed basis).

Ingredients, %	Normal Protein Basal Diet	Low-Protein Basal Diet	Analyzed Chemical Composition	Normal Protein Basal Diet	Low-Protein Basal Diet
Corn	57	58	Crude protein	18.8	17.2
Expended maize	5	5	Calculated DE, kcal/kg	3465	3436
Soybean meal (43% CP)	16	21	Dry matter	89.9	89.2
Soybean protein concentrate	6	0	Lysine	1.07	0.96
Rice bran meal	5	5	Calcium	0.63	0.62
Broken rice	5	5			
Fish meal	2	2			
Sucrose	1	1			
Calcium lactate	0.3	0.3			
Calcium hydrogen phosphate	1	1			
Limestone powder	0.1	0.1			
Trace mineral premix ^a^	0.1	0.1			
Vitamin premix ^b^	0.03	0.03			
Lysine (98%)	0.6	0.6			
Threonine	0.1	0.1			
Methionine	0.1	0.1			
Acidifier	0.4	0.4			
Antioxidants	0.15	0.15			
Choline chloride (50%)	0.12	0.12			

^a^ Providing the following amounts of minerals per kilogram on an as-fed basis: Zn (ZnO), 50 mg; Cu (CuSO_4_), 20 mg; Mn (MnO), 55 mg; Fe (FeSO_4_), 100 mg; I (KI), 1 mg; Co (CoSO_4_), 2 mg; Se (Na_2_SeO_3_), 0.3 mg. ^b^ Providing the following amounts of vitamins per kilogram on an as-fed basis: vitamin A, 8255 IU; vitamin D3, 2000 IU; vitamin E, 40 IU; vitamin B1, 2 mg; vitamin B2, 4 mg; pantothenic acid, 15 mg; vitamin B6, 10 mg; vitamin B12, 0.05 mg; nicotinic acid, 30 mg; folic acid, 2 mg; vitamin K3, 1.5 mg; biotin, 0.2 mg; choline chloride, 800 mg; and vitamin C, 100 mg.

**Table 3 animals-10-01852-t003:** The growth performance of piglets.

Items	NP	NC	PC	MCT	GML	*p*-Value
ADG (g/d)	145.23 ± 31.15	144.83 ± 60.76	153.96 ± 40.33	157.44 ± 56.86	145.62 ± 51.29	0.986
ADFI (g/d)	282.01 ± 27.76 ^b^	296.05 ± 41.47 ^b^	333.23 ± 30.39 ^ab^	355.64 ± 42.57 ^a^	335.70 ± 63.26 ^ab^	0.037
F/G	2.55 ± 0.72	2.49 ± 0.81	2.34 ± 0.87	2.31 ± 0.61	2.45 ± 0.59	0.979

Values are the mean ± SEM, *n* = 6 per treatment (the *n* = 6 refers to the number of pigs slaughtered and sampled). ^a,b^ Mean values sharing different superscripts within a row differ (*p* < 0.05). NP = normal protein basal diet no antibiotics included; NC = low-protein basal diet no antibiotics included; PC = low-protein basal diet + antibiotics (75 mg/kg quinocetone, 20 mg/kg virginiamycin and 50 mg/kg aureomycin); MCT = low-protein basal diet + 2 kg/T tricaprylin/tricaprin; GML = low-protein basal diet + 2 kg/T glycerol monolaurate; ADG = average daily weight gain; ADFI = average daily feed intake; F:G = feed:gain.

**Table 4 animals-10-01852-t004:** The intestinal permeability of piglets.

Items	NP	NC	PC	MCT	GML	*p*-Value
DAO (mmol/L)	2.18 ± 0.98	1.81 ± 0.71	2.06 ± 0.16	1.75 ± 0.49	1.68 ± 0.54	0.397
D-LACT (mmol/L)	11.63 ± 0.98 ^a^	9.90 ± 2.63 ^ab^	9.27 ± 2.26 ^b^	8.30 ± 1.33 ^b^	9.79 ± 0.64 ^ab^	0.041

Values are the mean ± SEM, *n* = 6 per treatment (the *n* = 6 refers to the number of pigs slaughtered and sampled). ^a,b^ Mean values sharing different superscripts within a row differ (*p* < 0.05). NP = normal protein basal diet no antibiotics included; NC = low-protein basal diet no antibiotics included; PC = low-protein basal diet + antibiotics (75 mg/kg quinocetone, 20 mg/kg virginiamycin and 50 mg/kg aureomycin); MCT = low-protein basal diet + 2 kg/T tricaprylin/tricaprin; GML = low-protein basal diet + 2 kg/T glycerol monolaurate.

**Table 5 animals-10-01852-t005:** The concentrations of cytokines in the jejunal mucosa of piglets.

Items	NP	NC	PC	MCT	GML	*p*-Value
SIgA (μg/mg)	10.90 ± 3.07 ^a^	8.42 ± 2.04 ^ab^	5.52 ± 0.93 ^b^	10.08 ± 1.62 ^a^	10.60 ± 1.27 ^a^	0.001
IL-1β (pg/mg)	173.12 ± 41.88	208.75 ± 34.17	214.32 ± 37.81	191.79 ± 30.95	183.28 ± 36.14	0.309
IL-6 (pg/mg)	109.38 ± 23.44	146.67 ± 43.86	139.86 ± 23.07	140.91 ± 29.18	136.59 ± 25.31	0.267
TNF-α (pg/mg)	32.28 ± 4.55 ^b^	45.22 ± 9.68 ^a^	42.54 ± 5.23 ^a^	44.00 ± 4.55 ^a^	43.77 ± 8.60 ^a^	0.020
IFN-γ (pg/mg)	9.54 ± 1.60	8.75 ± 1.66	9.45 ± 1.31	8.62 ± 1.18	8.83 ± 1.99	0.798

Values are the mean ± SEM, *n* = 6 per treatment (the *n* = 6 refers to the number of pigs slaughtered and sampled). ^a,b^ Mean values sharing different superscripts within a row differ (*p* < 0.05). NP = normal protein basal diet no antibiotics included; NC = low-protein basal diet no antibiotics included; PC = low-protein basal diet + antibiotics (75 mg/kg quinocetone, 20 mg/kg virginiamycin and 50 mg/kg aureomycin); MCT = low-protein basal diet + 2 kg/T tricaprylin/tricaprin; GML = low-protein basal diet + 2 kg/T glycerol monolaurate.

**Table 6 animals-10-01852-t006:** The concentrations of cytokines in the ileal mucosa of piglets.

Items	NP	NC	PC	MCT	GML	*p*-Value
SIgA (μg/mg)	12.47 ± 2.41 ^a^	9.9 ± 1.31 ^b^	9.33 ± 1.49 ^b^	12.95 ± 2.26 ^a^	14.26 ± 2.00 ^a^	0.001
IL-1β (pg/mg)	169.41 ± 30.75 ^b^	239.07 ± 30.05 ^a^	247.28 ± 10.66 ^a^	237.39 ± 11.16 ^a^	228.38 ± 42.09 ^a^	0.001
IL-6 (pg/mg)	123.27 ± 26.16 ^c^	218.48 ± 39.72 ^a^	190.34 ± 47.30 ^ab^	157.40 ± 33.76 ^bc^	180.23 ± 17.83 ^ab^	0.001
TNF-α (pg/mg)	38.72 ± 8.29 ^b^	60.79 ± 11.83 ^a^	61.91 ± 8.65 ^a^	56.59 ± 10.45 ^a^	52.27 ± 12.27 ^a^	0.006
IFN-γ (pg/mg)	9.15 ± 1.68	10.59 ± 1.15	10.86 ± 1.72	10.26 ± 1.83	11.56 ± 2.43	0.258

Values are the mean ± SEM, *n* = 6 per treatment (the *n* = 6 refers to the number of pigs slaughtered and sampled). ^a,b,c^ Mean values sharing different superscripts within a row differ (*p* < 0.05). NP = normal protein basal diet no antibiotics included; NC = low-protein basal diet no antibiotics included; PC = low-protein basal diet + antibiotics (75 mg/kg quinocetone, 20 mg/kg virginiamycin and 50 mg/kg aureomycin); MCT = low-protein basal diet + 2 kg/T tricaprylin/tricaprin; GML = low-protein basal diet + 2 kg/T glycerol monolaurate.

**Table 7 animals-10-01852-t007:** The serum concentrations of cytokines in the piglets.

Items	NP	NC	PC	MCT	GML	*p*-Value
SIgA (μg/mg)	30.34 ± 4.35 ^a^	20.03 ± 4.29 ^c^	19.47 ± 5.72 ^c^	22.36 ± 4.79 ^bc^	25.97 ± 2.23 ^ab^	0.010
IL-1β (pg/mg)	212.93 ± 25.52 ^b^	486.91 ± 81.45 ^a^	457.57 ± 119.31 ^a^	435.75 ± 53.86 ^a^	419.94 ± 36.03 ^a^	<0.001
IL-6 (pg/mg)	461.63 ± 83.95 ^c^	604.76 ± 29.53 ^ab^	619.55 ± 92.46 ^a^	512.28 ± 104.58 ^bc^	548.77 ± 66.47 ^abc^	0.014
TNF-α (pg/mg)	97.00 ± 24.34	120.91 ± 25.06	130.78 ± 19.88	106.99 ± 23.02	108.64 ± 16.79	0.136
IFN-γ (pg/mg)	33.78 ± 3.93	32.99 ± 3.11	37.89 ± 2.76	35.54 ± 4.11	32.06 ± 6.57	0.280

Values are the mean ± SEM, *n* = 6 per treatment (the *n* = 6 refers to the number of pigs slaughtered and sampled). ^a,b,c^ Mean values sharing different superscripts within a row differ (*p* < 0.05). NP = normal protein basal diet no antibiotics included; NC = low-protein basal diet no antibiotics included; PC = low-protein basal diet + antibiotics (75 mg/kg quinocetone, 20 mg/kg virginiamycin and 50 mg/kg aureomycin); MCT = low-protein basal diet + 2 kg/T tricaprylin/tricaprin; GML = low-protein basal diet + 2 kg/T glycerol monolaurate.

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
