# Peer review of "Low-Protein Diet Supplemented with Medium-Chain Fatty Acid Glycerides Improves the Growth Performance and Intestinal Function in Post-Weaning Piglets"

_animals, 2020, doi:10.3390/ani10101852_

Round 1

Reviewer 1 Report

The manuscript entitled “Low-protein diet supplemented with medium chain fatty acid glycerides improves the growth performance and intestinal function in post-weaning piglets” by Zhijuan Cui and co-workers presents the study on the influence of the supplementation of medium chain fatty acid glycerides to a low protein diet on the intestinal permeability, intestinal morphology and structure. The manuscript is prepared quite good, data are clearly presented and discussed. In my opinion, the presented work may be published in Animals after major revision. Some changes and additional discussion are needed.

My comments are presented below. Please provide the explanation for all of them, make changes in the text.

Major concerns:

- Introduction – page 2, lines 71-72 - Medium chain fatty acid glyceride is a monocarboxylic acid with 6-12 carbon atoms, including caproic acid (C6), caprylic acid (C8), decanoic acid (C10) and lauric acid (C12) – wrong definition is presented - medium chain fatty acid glyceride is not a monocarboxylic acid, is an ester formed from glycerol and fatty acid. According to the presented definition it suggests that the monocarboxylic acid with 6-12 carbon atoms, including caproic acid (C6), caprylic acid (C8), decanoic acid (C10) and lauric acid (C12) are strongly water soluble, but they are not.  

- Introduction – page 2, line 80 -  medium chain triacylglycerolshas – separate and correct– triacylglycerols have.

- Introduction – please indicate the validity of the performed study and selected animals

- Materials and methods – page 3, lines 97-98 - The animal experiments project identification code is IACUC#201302 and approved at 201906 – is the last part from the line 98 correct? It is a date?

- Materials and methods – page 3, line 118 – remove dot after ËšC

- Materials and methods – page 5, line 138 – correct the units

- Materials and methods – page 5, Immunohistochemical analysis – correct the description, it is not so correct and precise   

- Results, Intestinal permeability – how was the concentration of D-LACT determined

- Figure 3 – in my opinion it will be better to present three diagrams on one level next to each other, the same in the case of Figure 4

- Table 6 and 7 – expand column 1 to show the units or the item name in one line

- Abbreviations – include the ADFI, ADG, EDTA, PBS, F/G, SEM, BSA

- Use the reference style according to the scheme presented in instruction for authors, reference style section

- justify the text

Reviewer 2 Report

Major Comments

In general the manuscript requires major and extensive revision. Apart from the methods and results section, the introduction and discussion part do not support the findings and the scope of the study.  These sections should be rewritten. A backbone of the explanation of the results is missing in the discussion part.

Line 89: “on a low-protein diet that was benefit to relieve the nutritional burden” at the closing paragraph the aim of the study is presented in relation to the low-protein burden. The specific rationale was not explained previously in the introduction part, and therefore it is recommended to the authors to elaborate on the rationale of the study and explain by using also some reference why a low protein diet was implemented.

In the discussion part, extensive revision is recommended. The authors present a list of findings from several trials, doing an extensive literature review, but do not focus on the most important findings and how these could be explained.

Minor Comments

Line 29-30: define at first placement the abbreviations used such as “serum D-LACT”. Same remark for “ZO-1”. Please check and define all abbreviations at first use thoroughly

Lines 22-37: simple summary is written like a repetition of the abstract style. Perhaps the authors could get some ideas on the simple summary structure from some recent publications in order to revise it in a more reader friendly way

Line 65: change food to feed

Lines 78-79: revise sentence to make sense

Line 79: from this point new properties of medium chain fatty acids are listed. There should be a next paragraph inserted at this point

Line 110: correct collect to collected

Line 110: please define at the beginning of the paragraph when the blood and tissue collection took place. Presumably at day 14 of the experiment.

Lines 165-168: apparently mean values and statistical elaboration were performed by using each pen as the experimental unit (n=6 per treatment). Perhaps this should be mentioned in this section as in the Tables the n=6 is used and the readers should know what it stands for in each table. For example in the tissue analysis the n=6 refers to the number of pigs slaughtered and sampled, and not to the pens. Please make necessary additions to the Tables legends as well.

Line 171: change were to are

Lines 289-302: this paragraph could be shortened significantly and content should be moved to the introduction part (see also other comment)

Lines 303-313: this paragraph is a literature review with none or limited focus on the results of the study. It is recommended to revise or remove

Lines 325-338: this paragraph is a literature review with none or limited focus on the results of the study. It is recommended to revise or remove

Lines 350-351: “MCT significantly decreased the level of IL-6 in the ileum” compared to which treatment?

Round 2

Reviewer 1 Report

The revised version of the manuscript presented by Cui and co-authors entitled Low-protein diet supplemented with medium chain fatty acid glycerides improves the growth performance and intestinal function in post-weaning piglets", Authors gave responses for all of my questions and comments. After moderate English changes (minor revision), manuscript may be accepted for publication.

Author Response

Thank you for your reminding. We have revised the article in terms of language. Such as page 2, line 76, we have changed "has" to "have". 

Reviewer 2 Report

The authors have complied with the majority of the suggested changes.

Author Response

Thanks for your comment! We made a small number of language changes in the article. Such as page 5,line 166, we have changed 'was washed' to 'has washed'.